# Adapting Cities to Pluvial Flooding: The Case of Izmir (Türkiye)

**Stefano Salata** [1,*] , **Koray Velibeyoğlu** [2] , **Alper Baba** [3] , **Nicel Saygın** [2] , **Virginia Thompson Couch** [4] and **Taygun Uzelli** [5]

1   Department of Architecture and Urban Studies, Lab PPTE, Politecnico di Milano, 20133 Milano, Italy
2   Department of City and Regional Planning, Faculty of Architecture, Izmir Institute of Technology, Gülbahçe Campus Urla, Izmir 35430, Türkiye
3   Department of International Water Resources, Civil Engineering Faculty, Izmir Institute of Technology, Gülbahçe Campus Urla, Izmir 35430, Türkiye
4   Department of Architecture, Faculty of Architecture, Izmir Institute of Technology, Gülbahçe Campus Urla, Izmir 35430, Türkiye
5   Geothermal Energy Research and Application Center, Izmir Institute of Technology, Urla, Izmir 35430, Türkiye
*   Correspondence: stefano.salata@polimi.it

**Abstract:** In the coming decades, climate change will be one of the most significant challenges for urban areas. The quantity, duration and intensity of events, such as flash rains and heat waves, will increase the vulnerability of urban regions while exposing citizens to potentially dangerous conditions. According to the current literature, mainstreaming resilience in urban planning means designing rules that strengthen urban systems' adaptive and self-regulating functions by reducing their vulnerability. In this work, we aimed to build knowledge for the application of the sponge district concept to Izmir (Türkiye), one of Europe's most vulnerable areas to pluvial flooding. To do this, we first analyzed the runoff in each urban sub-watershed, then employed a composite index to determine potential areas of intervention for nature-based solutions. Results show that 10% of Izmir's urban areas are extremely vulnerable to cloudbursts, which means that 40% of the urban population is exposed to this phenomenon. Moreover, the runoff calculation in the sub-watershed demonstrated that the potential flood volume is underestimated, especially in the upslope areas. The results can be used as a template to suggest a stepwise approach to mainstream the resilience of densely-inhabited coastal urban catchments.

**Keywords:** pluvial flooding; ecosystem services; nature-based solutions; urban planning; resilience; vulnerability

## 1. Introduction

In the coming decades, climate change will be one of the greatest challenges for urban areas because the quantity, duration, and intensity of events, such as heat waves and flash rains, will increase the vulnerability of both natural and artificial systems, with an increased impact in urban areas due to the potential exposure of millions of citizens [1–3]. The concept of adaptation includes the possibility of thriving under these conditions while learning how to proactively and effectively respond to the changing dynamics [4,5]. Even the slippery concept of "resilience" deals with a system's ability to adapt [6]. In practical terms, the idea of resilience is operationalized to the empirical and spatial measurement of vulnerability and the identification of coping capacities in emerging dynamics [7,8].

According to previous works, mainstreaming resilience in urban planning means designing new rules that influence spatial planning processes and strengthening the adaptation and self-regulation of urban systems [9,10]. Mainstreaming resilience suggests that reducing vulnerability increases a system's capacity (and of all its components) to respond to turbulence and unexpected events and maintain or rapidly return to basic functions by

developing adaptive capacity and innovative solutions [11]. Izmir is extremely exposed to climate change hazards because of its impetuous growth, coastal geography, morphology, and density [12]. However, the urban planning approach to adaptation is still difficult to pursue, partly due to insufficient knowledge of the biophysical environmental condition of Izmir's urban system and partly due to a growth burst that Türkiye, in general, is experiencing [13]. While European cities are undergoing a shrinking population which leads to the reorganization of their internal space, the population of Türkiye is rapidly expanding [14,15]. Predictions for the Izmir metropolitan area foresee an additional inflow of almost 2 million citizens by 2030 [16]. This means a shift from 4.5 to 6.5 million in less than ten years.

With such extreme projected growth, a different urban planning approach is necessary to provide climate change adaptation in response to the coming challenge. Comprehensive urban planning should be developed according to the European Union guidelines for developing adaptation strategies at different levels, stated in brief as follows [17]:

- Build coherence within national and regional strategy on adaptation, avoid sectoral planning tools, and integrate the existing framework;
- Build knowledge of urban vulnerabilities at the local scale as an ordinary technical method to inform the decision-making phase;
- Promote and regulate the transition while providing local binding measures (land use zoning), from urban planning to environment and mobility.

Within these premises, this work focuses on the second of the points mentioned above while assuming that the knowledge of urban vulnerability is a pre-requisite to "promote and regulate the transition" (third point) to mainstream resilience. Therefore, this work focuses on mapping and assessing the vulnerability of Izmir's urban areas to "cloudburst events" [18]. According to Rosenzweig et al. (2019), three basic categories of knowledge support resilience to cloudburst: (1) knowledge of the contemporary weather and future climate conditions that determine cloudburst hazard, (2) knowledge of the vulnerability of urban socio-ecological systems and infrastructure systems, and (3) knowledge of potential strategies for cloudburst management.

Among many types of floods, here, we deal with the pluvial flood vulnerability modeled through conventional runoff calculation using integrated evaluation of ecosystem services and trade-offs (InVEST) [19]. As cloudbursts become more frequent and severe, cities will need to rapidly transform their structure while adopting the sponge districts' design and management principles [20–22]. To do so, a spatial assessment of the run-off is a basic pre-requisite for planning the adaptation with concrete and efficient supporting decision-making systems, which includes the design of capillary nature-based solutions (NBS) that increases the capacity of the drainage systems [23,24].

To integrate the runoff analysis, we studied the report of the metropolitan agency for water management (IZSU) for each urban watershed (sub-basin), we then evaluated if the predicted discharge capacity of urban streams is coherent with the quantity of runoff expected during a 70-mm cloudburst event (in the method section we detail why we select this threshold) [25,26]. The maximum discharge capacity of each artificial sub-basin was extracted and edited into a geographic information system geodatabase. This geoprocessing method aimed to discover if the discharge capacity predicted by IZSU in each river basin is sufficient to convey the potential runoff volume generated from a cloudburst event [27,28].

It is worth mentioning that runoff is an essential estimation of the vulnerability of an urban system. In very dense, coastal, hilly areas, runoff generates surface water flow paths that can be especially dangerous because soil sealing, canalization, and other alterations have modified the original stream network [29–31]. In addition, this research aims to create a bridge between knowledge of the system and the definition of management strategies, plans, or projects aimed at reducing the intrinsic vulnerability of an urban area [32,33].

In the proposed approach, the definition of areas vulnerable to cloudburst events is seen as the first step towards achieving "resilience of the system" and is intended as a process of cognitive reorganization of the system [34]. According to the vision of "co-

evolutionary resilience", resilience has become an interesting area of experimentation, especially concerning decision-making processes, because the re-definition of problems (e.g., vulnerability analysis) requires the reconstruction of common approaches used by the public administration, citizens, and stakeholders [35].

In this work, we demonstrate how the city of Izmir is subject to pluvial flooding vulnerability by using an auto-produced ancillary dataset that can be employed to map areas prone to flooding due to cloudburst events. The work is structured as follows: the materials and methods section presents Izmir's urban catchment and the runoff and vulnerability to cloudbursts that have been modeled in each watershed. In the results and discussion sections, the empirical outputs of the spatial model are first presented, then discussed in the light of possible NBS. Finally, the conclusion section briefly summarizes the main research questions and the achievement of results.

## 2. Materials and Methods

### 2.1. The Izmir Metropolitan Area

The Izmir Province spans 1,187,869.08 ha and is located on the western coast of Türkiye, creating a promontory in the Aegean Sea toward the Greek Archipelago (Chios, See Figure 1). Izmir has a distinctive identity: a heterogeneous topography, characteristic flora and fauna, a shape formed by the coast, vast natural resources, historical sites, and cultural heritage [36,37]. Izmir's altitude ranges between 0 and 1.200 m above sea level, and a Mediterranean climate with hot and dry summers and rainy winters characterizes the peninsula.

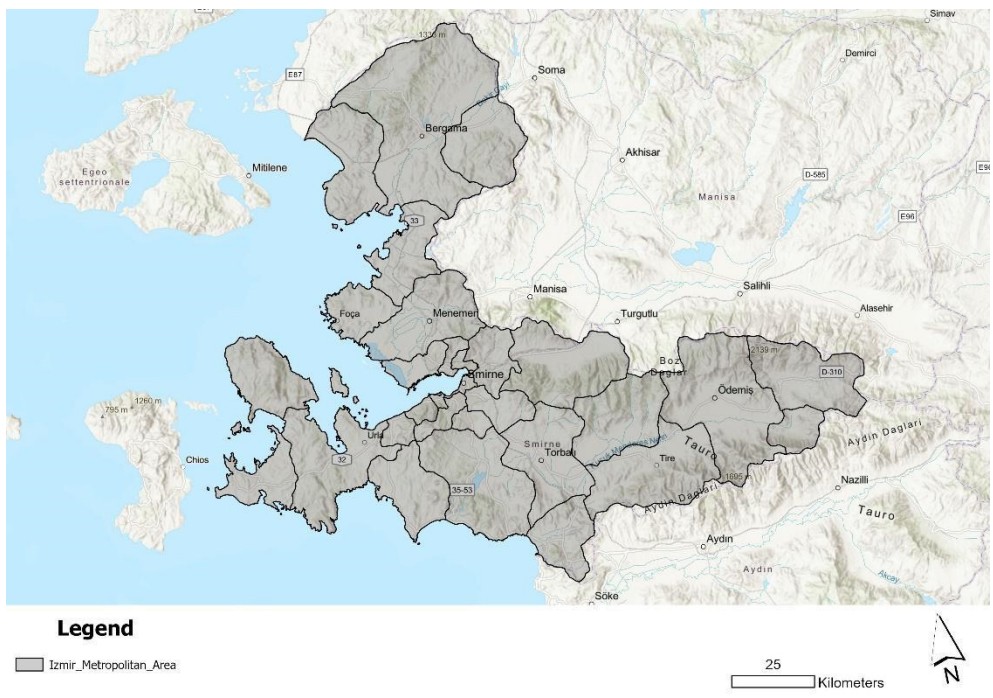

**Figure 1.** Izmir metropolitan area.

The Izmir province is experiencing one of Türkiye's most rapid urbanization processes. Data on the recent land-use change between 1990 and 2018 based on the Corine land cover dataset in the province of Izmir demonstrate that more than 33 thousand hectares of land were converted from agricultural or natural/semi-natural into urban uses while causing a marked reduction in the ecological integrity of this part of the Aegean promontory [13].

The urbanization process occurred at the expense of relatively flat, fertile agricultural areas (26,000 hectares). However, it is noticeable that the same process happened even at the expense of the characteristic natural and semi-natural Mediterranean environment

that is found in this part of Türkiye. More than 10,000 hectares of semi-natural land uses disappeared in the last 18 years, which reveals a strong biodiversity reduction process [12].

Another important characteristic to consider is the site-specificity of Izmir's urban land uses: the average soil sealing of the land-use class "continuous urban fabric" in the entire Izmir province is 75%, with peaks of 100% in some core districts [38]. Continuous urban fabric land uses are generally more sealed than the "industrial or commercial units" (44%). Comparatively, the "discontinuous urban fabric" land uses are sealed at 34%. Still, it should be considered that these land uses are largely sprawled and diffused in the catchment while defining the borders between the built and the unbuilt environment.

*2.2. Runoff Estimation*

The Urban Flood Mitigation model is a suite of the software InVEST (version 3.9.0), released by the Natural Capital Project. This model aims to verify the capacity of urban catchments to limit runoff and thereby avoid potential flooding in urban areas due to cloudburst events [20,23,39].

The Urban Flood Mitigation model considers the potential of porous green areas to reduce the runoff process by absorbing water, slowing surface flows, and creating space for water collection (in floodplains or basins) [40–42]. The model calculates the runoff reduction (i.e., the amount of runoff retained per pixel compared to the storm volume) and, for each watershed, provides an estimate of potential economic damage caused by flooding by overlaying information about potential flood extent and built infrastructure [20,43]. In this study, we employed the model without economic evaluation because of limited data and insufficient knowledge to correctly estimate the economic impacts. The model's algorithm assumes that flood-prone areas are those where there is a coupled interaction between permeable–impermeable materials of artificial surfaces (e.g., urban areas) and the quantity of soil drainage (depending on conductivity) that generates the amount of water that accumulates on the surface during a cloudburst, resulting in temporary flooding [21].

The required inputs are:

- Watershed vector, delineating areas of interest;
- Depth (in mm) of the rainfall of a single rain event;
- Land use land cover map (LULC);
- Hydrological group raster of the soil;
- The biophysical runoff curve value for each land use class in the land cover map.

Because runoff retention depends on the interaction between soil and land cover, the incorporation of the following two databases is crucial:

- Data on LULC is used to set the runoff curve number (RCN). The RCN represents the hydrologic soil capacity and indicates potential runoff (i.e., higher RCN indicates higher runoff potential).
- Data on hydraulic conductivity is used to establish soil drainage properties.

2.2.1. Depth of Rainfall

In this study, the session was launched using a depth rainfall of 70 mm as the modeling parameter. This value was chosen because it is far less than the values recorded in the rain event of 13–14 November 2020 and significantly more than 50 mm, considered the minimum threshold of "cloudburst" [44]. According to Türkiye's meteorological agency, Meteoroloji Genel Müdürlüğ (MGM), between 13 and 14 November 2020, 42.1 mm of rain fell in the town of Menderes during a single rain event. In Karabağlar, situated about 10 km to the north, rainfall of 77.3 mm was recorded. Elsewhere in the province, the service notified a depth between 147 mm (Urla) and 103.4 mm (Karaburun).

According to the IZSU water management report, a single rain event of 70 mm should have a return period of 200 years, thus constituting an extremely rare and exceptional event [45]. Unfortunately, between 2 and 3 February 2021 (less than four months later than

the November 2020 event), MGM reported that 123.9mm of rain fell in 24 h in Konak district of İzmir, while on 03 February, a rainfall of 130.9 mm was observed in Menderes district.

### 2.2.2. Land Use Land Cover

Land use land cover has been built entirely around the RCN classification. Runoff curve numbers A, B, C, and D were associated with classes of urban soil and their permeability. The land use classification is based on the USDA classes [27,46,47], by employing the "imperviousness" high-resolution layer (HRL) database available at https://land.copernicus.eu/pan-european/high-resolution-layers/imperviousness accessed on 25 July 2022 [48,49].

The imperviousness products capture the percentage and rate of change of soil sealing [50]. The imperviousness HRL represents the spatial distribution of artificially sealed areas, including the degree of soil sealing per unit area. The level of soil sealing (degree of imperviousness from 1 to 100) is produced using a semi-automated classification based on calibrated normalized difference vegetation index [51–53]. The forest density dataset was then used to classify the unbuilt land. The tree cover density (TCD) product consists of status layers showing the tree cover percentage, which is available for 2012, 2015, and 2018 reference years [54]. The tree cover change mask (TCCM) product was used to reveal the change in tree masks for 2012–2015 and 2015–2018. This product is available on the Land Copernicus website under the Tree Cover Density products domain. The imperviousness and forest datasets were thus combined to obtain a land use classification based on the runoff curve numbers of both built and unbuilt land.

Geographic information system (GIS) data processing was conducted using ESRI ArcGIS software (ver.10.8.1, Redlands, CA, USA) licensed by the Izmir Institute of Technology. Using ESRI ArcGIS a reclassification sampling and raster combination have been performed. This was done in three steps:

1.  a discrete classification of the urban areas, beginning with a continuous imperviousness value;
2.  a discrete reclassification of the unpaved areas, beginning with a continuous forest value and employing three classes (poor, fair, good); and
3.  a utilization of the raster combine tool to reach the final classification.

### 2.2.3. Soil Hydraulic Conductivity

The second database represents the saturated hydraulic conductivity (Ksat, mm/h), defined as the soil's ability to be vertically drained of liquids when it is in a saturated condition [23,55]. Soil with high conductivity will allow water absorption and movement within a short period. Water quickly reaches the aquifer and surface flow processes are limited. Conductive soils are thus protected against surface erosion and ensure better surface water quality [44]. However, because of their high conductivity, permeable soils do not protect groundwater. In contrast, where there is low infiltration, there is more surface runoff.

Hydraulic conductivity is a function of soil porosity: the water's movement is facilitated when pores are large and continuous, while it is made more difficult when pores are small and disconnected [24]. A soil's porosity is related to its texture: clayey soils generally have a lower saturated hydraulic conductivity than sandy or gravelly soils, in which pores are less numerous but larger, and facilitate the passage of a significant volume of water [56,57].

We created an ancillary dataset that is based on the more accurate geological unit definition found in the geological map of Izmir [58,59]. Classification has been made using a specific hydrological soil classification (see Table 1). All spatial inputs of the model are published as Supplementary Materials.

**Table 1.** Hydrological soil group classification of geological units around İzmir.

| Geological Units | Hydrological Soil Groups | Infiltration Rate ** | Runoff Potential ** |
|---|---|---|---|
| Recent alluvium deposits | A | High | Very Low |
| Continental clastics | B | Moderate | Low |
| Volcano-sedimentary units | C | Slow | Moderate |
| Volcanic units (andesite, dasite) * | C | Slow | Moderate |
| Flysch | D | Very Slow | High |
| Carbonates (Miocene) | B | Moderate | Low |
| Carbonates (Cretaceous–Flysch Blocks) * | B | Moderate | Low |
| Carbonates (Cretaceous–Jurassic) * | C | Slow | Moderate |

* These units generally spread out as rock mass in the field and have been included in the classification considering the possibility of creating altered and degraded soils. ** The comparative correlation of the units generally determines these criteria.

To better understand the spatial distribution of the modeling output, the runoff index has been analyzed using the official subdivision of Izmir's urban water basins, which IZSU prepared. These watershed subdivisions of the city's stream network have been superimposed on the runoff map. All of the sub-basins were georeferenced, including the polylines of the streams, to obtain a georeferenced environment of the official sub-basins from the original Autocad.dwg files. We transformed all runoff data and discharge capacity into cubic meters/second.

To detect potential deficit zones, we then compared the peak flow capacity for a return time of 200 years (based on the IZSU report on the artificial basins, which considers the frequency of a rain event of 70 mm to be around 200 years) with the runoff modeled by InVEST. The following are the volumes of water predicted for each watershed (Q) that we entered into our geodatabase: Q1, 21 $m^3/s$; Q2, 8 $m^3/s$; Q3, 12 $m^3/s$; Q4, 8 $m^3/s$; Q5, 2.8 $m^3/s$; Q6, 3.7 $m^3/s$; Q7, 12 $m^3/s$; Q8, 14 $m^3/s$; Q9, 13 $m^3/s$; Q10, 7.5 $m^3/s$; Q11, 16 $m^3/s$; Q12, 20 $m^3/s$; Q13, 2.5 $m^3/s$; Q14, 5 $m^3/s$; Q15, 26 $m^3/s$; Q16, 25 $m^3/s$; Q17, 150 $m^3/s$; Q18, 65 $m^3/s$; Q19, 4.5 $m^3/s$; Q 20, 80 $m^3/s$; Q21, 85 $m^3/s$; Q22, 100 $m^3/s$; and Q23, 140 $m^3/s$.

### 2.3. Composite Vulnerability Index

As anticipated, although runoff is a basic indicator used in calculating the biophysical absorption capacity of urban catchments, a more detailed integration is needed to assess the pluvial flooding vulnerability of sloped coastal areas such as the urban catchments of Izmir [37,60]. The most serious problem in these areas is that once runoff is generated, it moves on the urban surface, forming critical streams and debris flows that are dangerous for citizens [61]. To address this problem, our analysis of Izmir's urban streams has integrated the biophysical results of runoff. The underlying assumption is that the run-off generated in coastal cities that were developed on sloped areas is potentially more problematic than that of flatter areas [11,30]. On a sloped area, the more runoff is produced, the greater the chance it will transform into an urban stream with dangerous consequences. Unfortunately, Izmir is built on steep and high slopes, creating a massive problem of water flowing downhill to the coast during flash rains. Stream analysis has been performed using the hydrological processing toolbox of ESRI ArcGIS and using a digital elevation model of Izmir as input to produce the direction of the stream and flow accumulation. Then, the pluvial flooding vulnerability was calculated with a raster overlay which outputs a composite per-pixel index that sums the runoff and flow accumulation raster. Once calculated, the index was normalized as 0–1. When calculated as a simple raster sum, the pixel value tends to 1 when high runoff production is associated with a huge flow accumulation. On the contrary, where the runoff is insignificant and the flow accumulation is low, there is no vulnerability (the pixel value is 0).

The composite index can be viewed as an early monitoring system that can determine if any areas will be vulnerable during rain events of 70 mm or more.

To obtain a tabular sum of the vulnerability index for each district, we intersected the vulnerability raster with the "Urban Atlas" database (2018), a highly detailed land use dataset containing population data for each city block and that is available on the EU Copernicus website [30,62]. By this method, we also calculated the people's exposure to flood phenomena (intended as the population living in vulnerable areas) [63,64]. To obtain a clearer analytical assessment, we additionally made a second geoprocessing intersect with soil type and created a pivot table, which we used to interpret our results. We calculated the average value of vulnerability for each block, for each land use class and soil type, using population density, imperviousness, and forest density. We then observed the relationships between these parameters and vulnerability to cloudburst.

## 3. Results

The runoff retention index elaborated by InVEST for each district of Izmir is reported in Table 2. The index ranges from 0 (no retention) to 1 (full retention) and expresses the infiltration capacity rate for each district on the rain volume. The retention is also expressed in cubic meters (we used this parameter for the analysis in each sub-basin, see Section 3.1) while, in the third column, there is the runoff volume (flood volume), also measured in cubic meters.

**Table 2.** Runoff calculation in each district.

| District | Runoff Retention Index | Runoff Retention (m$^3$) | Flood Volume (m$^3$) |
|---|---|---|---|
| Cigli | 0.84 | 8,158,642.22 | 1,521,019.73 |
| Aliaga | 0.69 | 18,105,906.11 | 8,004,562.92 |
| Balcova | 0.81 | 820,531.22 | 196,638.80 |
| Bayindir | 0.81 | 30,928,661.83 | 7,424,428.69 |
| Bayrakli | 0.57 | 1,191,601.97 | 899,942.11 |
| Bergama | 0.73 | 78,860,700.05 | 29,078,878.64 |
| Beydag | 0.68 | 8,178,795.84 | 3,845,999.70 |
| Bornova | 0.71 | 10,962,628.64 | 4,443,664.73 |
| Buca | 0.70 | 8,772,675.44 | 3,685,273.85 |
| Dikili | 0.72 | 26,694,200.35 | 10,297,033.60 |
| Odemis | 0.75 | 53,281,697.82 | 17,973,245.00 |
| Cesme | 0.64 | 12,358,222.64 | 6,804,634.84 |
| Foca | 0.68 | 11,731,502.51 | 5,468,875.75 |
| Gaziemir | 0.73 | 3,619,550.70 | 1,317,360.30 |
| Guzelbahce | 0.77 | 4,145,279.71 | 1,262,850.53 |
| Kiraz | 0.66 | 26,375,302.33 | 13,774,373.22 |
| Karsiyaka | 0.62 | 2,179,324.87 | 1,337,664.15 |
| Karabaglar | 0.65 | 4,066,285.30 | 2,164,219.08 |
| Karaburun | 0.76 | 21,892,952.08 | 6,916,458.82 |
| Kemalpasa | 0.86 | 41,048,607.24 | 6,651,606.75 |
| Kinik | 0.76 | 25,544,893.69 | 7,967,179.04 |
| Konak | 0.46 | 730,074.96 | 866,072.15 |
| Menderes | 0.82 | 44,775,023.18 | 9,555,442.95 |
| Menemen | 0.82 | 32,537,933.40 | 7,368,420.30 |
| Narlidere | 0.79 | 2,738,442.20 | 714,846.99 |
| Seferihisar | 0.74 | 20,316,627.62 | 7,096,493.69 |
| Selcuk | 0.81 | 17,855,982.98 | 4,267,684.92 |
| Tire | 0.76 | 38,225,117.63 | 11,929,215.41 |
| Torbali | 0.87 | 35,282,145.22 | 5,098,592.61 |
| Urla | 0.76 | 37,185,884.41 | 11,618,872.87 |

The table shows that the mean retention index in the metropolitan area is 0.73 (min. 0.46 in Konak and max 0.87 in Torbali) with a total retention capacity of 628,565,194.2 cubic meters and a potential flood volume of 199,551,552.2 cubic meters. Notably, there is huge variability in the runoff retention indexes for the districts that compose the metropolitan

area of Izmir. These differences rely on the heterogeneity of the hydraulic soil conductivity within the catchment. Runoff retention is much higher in the plain–fertile districts where sediments are present, and the ground can more efficiently absorb water, while the upslope areas, where the most recent urban expansions are concentrated, show poor conductibility. A major problem arises when the dense settlement system is built on poorly conducting soil. The municipality data show the worst performance in the Konak district, with a runoff retention index of 46%. Poor runoff retention also affects Bayrakli, Karsiyaka, Cesme and Kiraz, where the index does not exceed 65% of the rain volume. In these specific districts, there is a high probability that a cloudburst event will result in temporary urban floods with potential debris flows.

### 3.1. Runoff Analysis of Each Sub-Basin

As anticipated, we used the original InVEST output to verify if the modeled runoff quantity is higher or lesser than the sub-basin water discharge capacity assessed by the IZSU technical reports. We basically verified if, during a single rain event of 70 mm, there is sufficient or insufficient discharge capacity in each sub-basin of the city of Izmir (Figure 2). We thought that this analysis (which we labeled as deficit analysis) can be relevant, especially to measure the rainwater harvesting system in the new urbanization along the urban fringe in the uphill areas where the natural streams are intercepted, deviated, and rectified by channels and pipes.

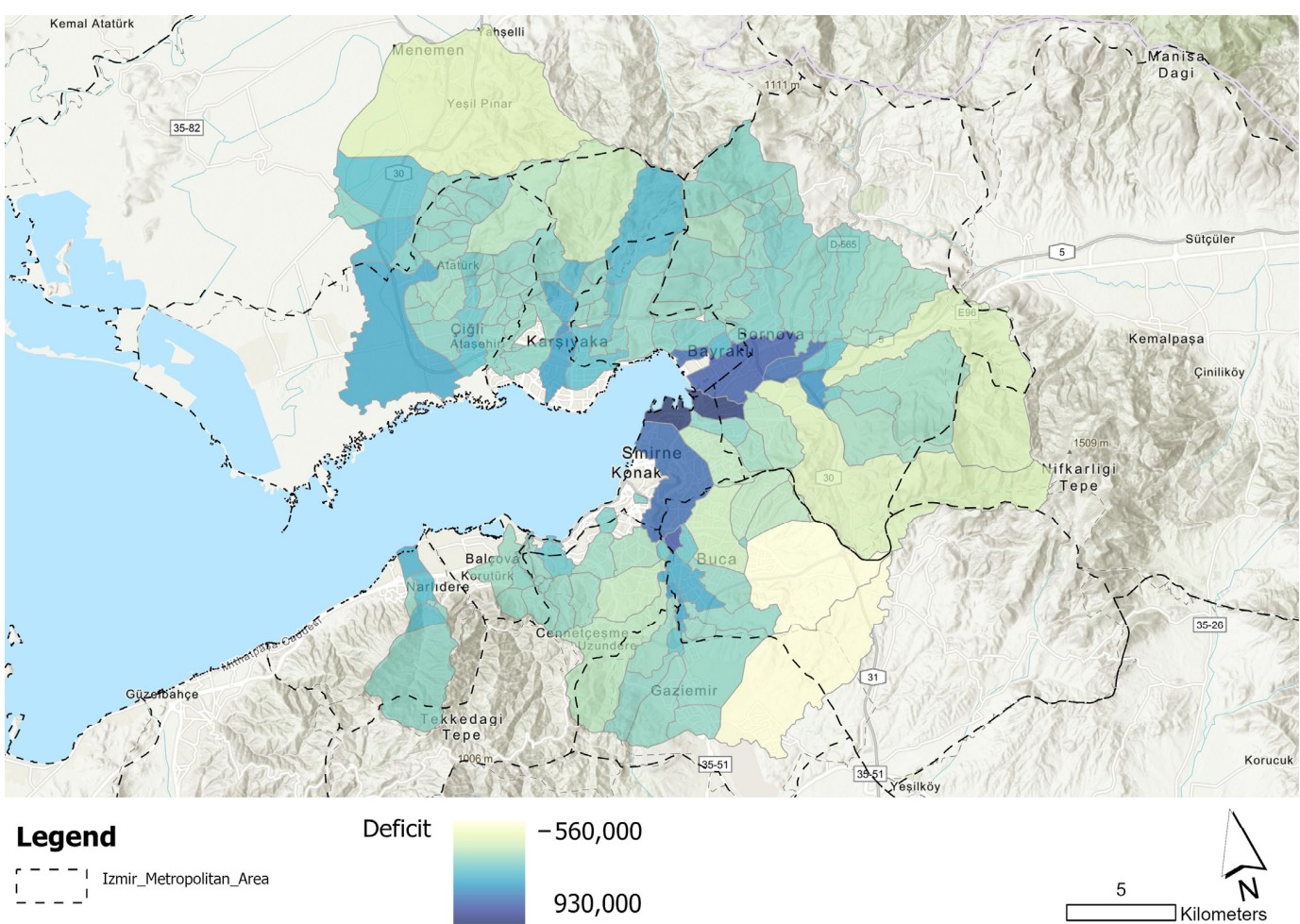

**Figure 2.** Flood volume deficit analysis of the urban watershed (cubic meters).

In this final map of the urban districts of Izmir, the light-yellow colors represent urban water basins that seem to present a deficit in harvesting capacity, according to the runoff

modeling quantification. This means the natural or artificial stream bed is not adequate to discharge the flood volume simulated by our model during a 70-mm cloudburst. The areas of the map colored blue are those in which the rainwater harvesting network seems sufficient to discharge the runoff quantity.

Our modeling covered 10 watersheds in the densely inhabited part of the metropolitan area (out of 23), with a further subdivision of 158 sub-watersheds which composes the entire urban catchment (see Table 3). The estimated cumulative flood volume during a 70 mm cloudburst calculated by our model in these sub-watersheds is of 13,307,225.9 cubic meters, while the total discharge capacity for the same quantity of rain is 23,270,076.0. This means that, in the entire catchment, the discharge volume seems to be enough to gather all the runoff (urban canals and streams are sized for this kind of rain). The deficit analysis shows that, according to IZSU's report, 9,962,850.1 additional cubic meters can be discharged by the system.

**Table 3.** Discharge capacity deficit for each watershed.

| Watershed | Runoff Volume ($m^3$) | Max Discharge Volume ($m^3$) | Deficit ($m^3$) |
|---|---|---|---|
| 1 | 1,444,957.37 | 2,113,200.00 | 668,242.63 |
| 2 | 1,137,010.88 | 1,726,200.00 | 589,189.12 |
| 3 | 1,570,709.80 | 3,839,400.00 | 2,268,690.20 |
| 4 | 50,111.03 | 90,000.00 | 39,888.97 |
| 5 | 1,435,246.14 | 2,595,600.00 | 1,160,353.86 |
| 6 | 2,269,338.48 | 3,546,000.00 | 1,276,661.52 |
| 7 | 4,610,108.23 | 7,701,120.00 | 3,091,011.77 |
| 8 | - | 122,400.00 | 122,400.00 |
| 9 | 520,722.53 | 579,240.00 | 58,517.47 |
| 11 | 269,021.45 | 956,916.00 | 687,894.55 |
| Tot | 13,307,225.92 | 23,270,076.00 | 9,962,850.08 |

However, the situation is different if we concentrate the analysis on a single sub-watershed. There, the deficit analysis shows some deficits: 59 sub-watersheds out of 158 show some deficits, meaning that their potential discharge capacity is less than the modeled flood volume.

The map shows that the deficit area is mainly concentrated in the "upstream" part of the metropolitan system. In these areas, the expected discharge peak flow of the artificial hydraulic network is insufficient to convey the simulated runoff volume. This means that in these sub-watersheds, there may be an underestimation of the flood dynamics induced by the sealed surfaces or an overestimation of the potential soil hydraulic conductivity capacity in the naturally unsealed areas. At the same time, these upstream basins are undergoing a process of extreme anthropization and land-use change due to a huge expansion of urban settlement.

The deficit analysis indicates that Bayrakli and Karşiyaka are two of the problematic districts that need to be carefully studied. Among the many, these two districts appear to be problematic both for their limited runoff retention capacity and for their potential deficit in rainwater harvesting capacity while becoming extremely suitable to host specific NBS aimed at increasing water absorption [65–67].

### 3.2. The Vulnerability of Urban Areas

The vulnerability distribution is concentrated not only in the dense central part of Izmir (Konak, Karabaglar, Karsiyaka, Bornova, and Buca), but also in the far eastern districts of Kiraz and Beydag, in the western peninsula of Cesme, and in the northern districts of Foca and Aliaga.

The output (Table 4) represents the calculation of the average vulnerability of the pixels in each urban district of Izmir. This classification helps to expand the potential areas of intervention where designing NBS is a must.

**Table 4.** Vulnerability analysis for each land use typology.

| Land Use/Land Cover | Average Vulnerability (Index 0–1) | Average Block Dimension (sq. m.) | Average Imperviousness (%) | Average Forest (%) | Pop/ha |
|---|---|---|---|---|---|
| Airports | 0.20 | 1,022,491.82 | 30.09 | 2.58 | - |
| Arable land (annual crops) | 0.16 | 312,838.00 | 2.51 | 10.16 | 2.01 |
| Complex and mixed cultivation patterns | 0.25 | 93,845.99 | 1.06 | 20.22 | - |
| Construction sites | 0.33 | 35,815.19 | 20.76 | 0.82 | - |
| Continuous urban fabric (S.L.: >80%) | 0.78 | 4,244.60 | 89.29 | 0.37 | 350.79 |
| Discontinuous dense urban fabric (S.L.: 50–80%) | 0.56 | 8,846.49 | 61.42 | 3.08 | 211.05 |
| Discontinuous low density urban fabric (S.L.: 10–30%) | 0.29 | 13,878.40 | 25.00 | 6.51 | 60.20 |
| Discontinuous medium density urban fabric (S.L.: 30–50%) | 0.39 | 12,887.26 | 40.88 | 5.40 | 98.06 |
| Discontinuous very low density urban fabric (S.L.: <10%) | 0.22 | 10,004.88 | 9.83 | 8.21 | 30.15 |
| Fast transit roads and associated land | 0.36 | 181,461.71 | 34.48 | 1.43 | - |
| Forests | 0.21 | 342,492.43 | 0.16 | 56.22 | - |
| Green urban areas | 0.32 | 14,102.58 | 20.66 | 19.10 | - |
| Herbaceous vegetation associations (natural grassland, moors...) | 0.26 | 311,558.56 | 2.24 | 29.78 | - |
| Industrial, commercial, public, military and private units | 0.40 | 21,761.79 | 43.44 | 3.32 | 28.88 |
| Isolated structures | 0.15 | 4,728.27 | 2.28 | 14.89 | 48.16 |
| Land without current use | 0.37 | 10,960.48 | 28.06 | 4.50 | - |
| Mineral extraction and dump sites | 0.31 | 52,556.34 | 11.15 | 2.20 | - |
| Open spaces with little or no vegetation (beaches, dunes, bare rocks, glaciers) | 0.20 | 64,715.20 | 2.04 | 6.25 | - |
| Other roads and associated land | 0.40 | 262,561.67 | 16.67 | 10.23 | - |
| Pastures | 0.23 | 67,119.48 | 1.51 | 10.59 | 3.85 |
| Permanent crops (vineyards, fruit trees, olive groves) | 0.17 | 144,370.54 | 1.02 | 22.53 | 1.85 |
| Port areas | 0.47 | 11,270.88 | 48.10 | 3.12 | 681.99 |
| Railways and associated land | 0.37 | 63,622.48 | 40.75 | 2.95 | - |
| Sports and leisure facilities | 0.35 | 25,422.58 | 29.90 | 9.14 | 14.15 |
| Water | 0.18 | 238,742.25 | 9.65 | 9.47 | - |
| Wetlands | 0.12 | 449,345.63 | 1.78 | 10.46 | - |

According to Table 4, the average hydraulic vulnerability value has been evaluated using the major land use categorization. This categorization allows us to see which land use typologies are more problematic in the city while furnishing an indication of which policy recommendation can be more effective in adapting the urban system to climate change.

The average vulnerability within the continuous urban fabric is 0.78, with an average block size of 4,244.60 sq. m., an average rate of imperviousness of 89.29% of the plot surface, an average forest cover area of 37%, and an average population density of 350.79 persons/ha. This land-use typology is one of the most critical since it is where the vulnerability is concentrated on densely inhabited, historical and fragmented settlements where the potential application of NBS can be problematic and difficult to realize.

Nevertheless, if reducing the hydraulic risk is a goal, the typical existing continuous urban fabric should be considered for minor, capillary, or de-sealing interventions (see Figure 3).

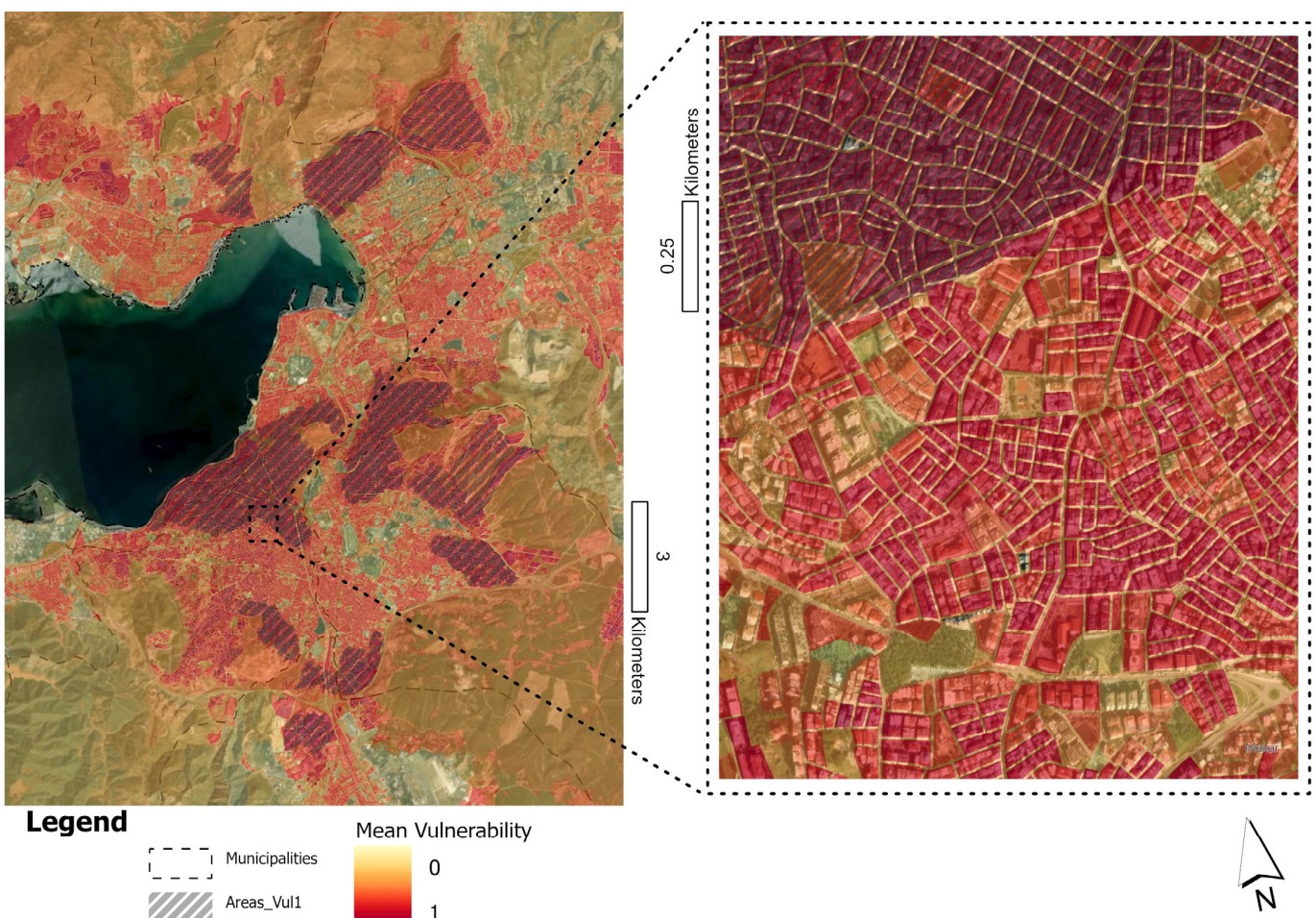

**Figure 3.** Vulnerability analysis of urban blocks.

According to the urban atlas dataset, the anthropic areas of Izmir (macro class 1) span 57.170 ha and have a population of 3,818.232. To better evaluate the results of our vulnerability analysis, we made a further final step: an evaluation of the most vulnerable areas using a 99% confidence interval. We discovered that 5.717 ha (10%) of the above-mentioned anthropic areas could be considered extremely vulnerable. This means that in these densely inhabited central areas, there is a combined action of high runoff generation plus huge downstream flows due to their slope. An exposure analysis reveals that 40% of the total population of Izmir's anthropic areas lives in these extremely vulnerable areas (1,525,727 citizens). It should be noted that the analysis of vulnerable areas does not necessarily overlap with the most commonly used map of the floodable areas (the plain areas of riverine flooding). This is because they represent the parts of the land where the run-off is generated before it moves for morphological characterization (not where the flood is stagnating).

## 4. Discussion

### 4.1. A Stepwise Approach to Adaptation

Adding new urban areas in a peri-urban fringe without a clearly demonstrated capability to estimate the effects of these new areas on cumulative runoff can be dangerous for the entire urban system and expose more of the population to the problem of cloudburst flooding. Furthermore, undersizing the artificial rainwater harvesting system in the newly urbanized upstream area can contribute to generating problems in plain areas downstream, which already experience high exposure to flooding.

The analysis of the riverine peak flow capacity for each sub-watershed demonstrated to what degree the runoff volume is underestimated in the upslope areas. Results also show the delicate integration and connection between the natural/meandric and anthropic/rectified stream networks: in the upstream areas, high retention semi-natural basins should be used to intercept the flows before they enter the densely urbanized system, to reduce inflows and mitigate the impact of runoff during cloudburst events.

Nature-based solutions are a reasonable option for managing stormwater in Izmir. The amount of permeable open spaces within the city is far less than it should be. Therefore, the existing natural open spaces of the city should be protected and improved to increase the porosity of these areas, and the impervious area should be gradually reduced by permeabilization and afforestation projects.

In the plains, the impacts of flood events can be prevented by upstream measures, such as slowing the flow (detention) of runoff, capturing water upstream in cisterns (retention) and increasing the permanent vegetation. In addition, new construction in uphill areas should not be allowed without an ex-ante evaluation of the soil's hydraulic characteristics and an estimation of the potential runoff [27].

To slow the runoff and reduce it, NBS can be realized, such as renaturalization of stream beds, construction of floodable parks, soil enrichment, creation of bioswales and rain gardens, and other methods of reducing the area of impervious surface and of harvesting rainwater in both existing urban areas and new developments [68,69].

These NBS can provide additional benefits, such as reducing water pollution in Izmir's streams and bay, recharging aquifers which are the primary freshwater resource of the city and which are currently at very low levels, reducing the heat island effect, increasing biodiversity, and reducing car dependence by providing pedestrian and slow-mobility connections along natural stream corridors between the bay and the urban fringe. All of these together would provide an overall improvement in the quality of the living environment of the city.

Some NBS are selected to maximize biodiversity benefits and deliver the key objectives of managing flood risk and water quality in urban areas. Hereafter, we only discussed the kind of solutions and approaches that benefit runoff control, dividing the measure adopted at the comprehensive city level by the type of solutions that can be designed and delivered at the district level.

At the comprehensive city level, Izmir's metropolitan plan is already implementing the living park project, which promotes peri-urban open space preservation through land protection [38,70]. This project aims at connecting the peri-urban inland areas with the historic waterfront Park. Some districts are also experimenting with green infrastructure applications based on ecosystem mapping and designing the afforestation of urban areas while maximizing urban biodiversity. Unfortunately, according to our analysis, the above-mentioned experimental measures can reach only a partial result if they are not envisioned in a comprehensive approach that introduces some concrete steps toward adaptation. One of the most important results of our analysis indicates that citizens' exposure to temporary flooding can be reduced only by identifying suitable areas to intercept the upstream flow by detention ponds and basins. While small ponds can be located in urban catchments, finding the proper space for new detention basins (dams) is much more problematic (geomorphological characteristics of suitability).

As for the district level, Izmir is still underestimating the importance of de-paving the city both in public and private areas while adopting strict urban regulation measures that introduce bioswales, rain gardens, green streets, green roofs, and green parking lots.

As shown in Figure 4, Izmir's pathway toward adaptation can be discredited in six steps. These steps were designed as a sample for a general guideline design approach on Izmir's catchment while targeting the main problems outlined in the results of this research that are hereafter synthesized: (1) the upstream run-off is generally underestimated in relation to the urban discharge capacity of canals; (2) the retention capacity is higher in flat sediment areas (which are just a small portion of the city); (3) the process of anthropization

should be stopped to allow for analyses of the flow routes and to respect the streambeds; (4) the existent stock of built-up land should be progressively depaved by re-naturing the city; (5) the forest cover should be increased where possible; and (6) the stream flows should be intercepted by detention ponds before they enter into the densely inhabited area.

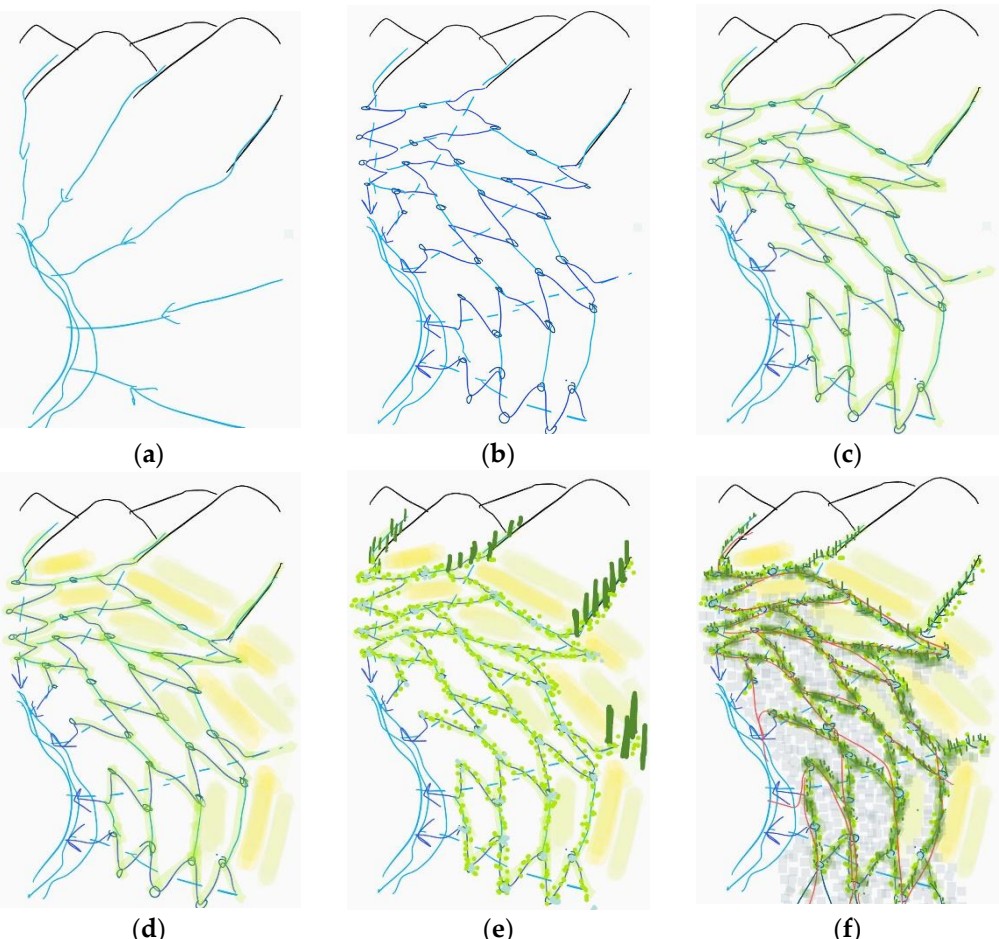

**Figure 4.** A step-by-step approach to adaptation to flooding in Izmir. (**a**) Identifying the stream routes. (**b**) Elongating the stream routes and intercepting water with micro detention ponds. (**c**) Planting autochthonous vegetation. (**d**) Promoting agroforestry in peri-urban upslope areas. (**e**) Afforestation of streambeds in the mountain. (**f**) Greening the walking and bike paths in Urban Areas.

Considering these points, we suggest some policy recommendations that are designed to cope with the mentioned elements. First, the streambed's protection is an absolute priority (Figure 4a). Even though the streams are hardly rectified and heavily sealed in the urban catchment, their maintenance and protection from new urbanization are a must. Therefore, the identification of the stream routes should be used to avoid potential interferences between the new urban transformation and the natural water basin.

The second move (Figure 4b) should consider a diffuse, capillary, and meandric intercepting water system that is mainly designed and realized using existing public spaces (roads, pedestrians, plazas) [40,71,72]. Here, the replacement of concrete from the topsoil with small strips of permeable land can play a significant role in intercepting the surface runoff while steering the flows to small detention ponds (which can be on public open green spaces, private rain gardens, water harvesting parks or green filter strips). This newly designed intervention should be accompanied by re-vegetation with autochthonous shrubs and trees (third move, Figure 4c).

A further step (fourth move, Figure 4d) is the promotion and valorization of a diffuse agro-afforestation practice in the peri-urban land surrounding the dense and compact

settlement system [73,74]. The utilization of a more sustainable, less hydro-demanding, terraced system can be specifically designed to implement anti-erosive practices and used to intercept the rainwater in densely vegetated areas. Finally, two greening procedures should be strongly recommended and organically included in the city's plan (moves five and six): the afforestation of the streambeds in the mountainous areas (Figure 4e) is necessary to reduce rainwater flood volume before it enters the urban catchment, while the greening of the capillary interceptor meandric system (Figure 4f) is a measure to augment the infiltration capacity while augmenting urban biodiversity.

### 4.2. Some Additional Rules

Urban planners and architects design urban transformation considering first the built-up space, while green areas result from the "voids" between the buildings and the streets. The concrete application of NBS requires a change in the paradigm: green areas and multifunctional spaces should be designed according to the sponge district concept, following watercourses and defining the water catchments. Even the disposition of trees and vegetation should be designed considering the crucial function of the permeable surface and vegetation on the soil hydraulic dynamic [23,75].

According to our findings, the preliminary analysis of the natural drainage pattern at the site of intervention should be evaluated by detailed spatial characterization of the existing flow paths and the discharge points. The utilization of high-resolution Digital Elevation Models and Digital Terrain Models can be integrated by LIDAR techniques and ancillary photo-interpretation based on Unmanned Aerial Vehicles. In this phase, soil scientists, hydrologists, and ecologists should work hand in hand to characterize the site's flow path, morphology, and topography. Historical drainage stream maps can be used to understand the historical process of urbanization.

A micro-zoning of potential infiltration capacity can be extremely helpful in understanding where to dispose of the green areas to convey water runoff. In this phase, the goal is to design green, permeable areas to intercept the streams.

Interception micro-zoning and flow routes should be used to define sub-catchments and divide the plots where urbanization will be concentrated by the plots where open public spaces can be placed, particularly, on larger sites. Public and private green permeable areas can host detention ponds and convey downstream water flow to the discharge points. Where appropriate, planned parks and open spaces should be located at the downstream end of sub-catchments to provide space for larger-scale surface water attenuation and controls.

### 4.3. Limits and Potentialities

The model we employed for this study simplifies the process of runoff discharge by considering that water on impervious surfaces moves directly to adjacent areas, contributing directly to surface flow accumulation. Nevertheless, in dense urban catchments, building roofs, terraces, and other horizontal or vertical surfaces that receive rainwater do not directly contribute to soil runoff. Instead, the water is temporarily contained in various paths (e.g., the pipes of the stormwater sewer system), contributing to the total discharge sometime later (downstream). Specific literature on this dynamic demonstrates how the biophysical quantification of the runoff in the built environment exposed to torrential hazards can be difficult to estimate because a multitude of factors, such as the quantity, quality, and surface of buildings, the sewer system, the soil type, and the dryness of the soil, can affect discharge volume during a cloudburst. Nevertheless, the new InVEST flooding risk mitigation model has been augmented with the ability to also include in the run-off calculation the evapotranspiration and interception by vegetation or the built environment. Thus, an improvement of this study would consider a replication of the vulnerability analysis while using the recently updated InVEST release.

Additionally, the soil's conditions before the cloudburst event can play a significant role in changing the infiltration capacities: after long dry seasons, the first rain event can

be extremely dangerous because the dust deposition on the ground can be "compacted" by the rain before eventually becoming permeable. In this case, the runoff volume can change significantly.

Therefore, the results of this study should be evaluated, bearing in mind that the InVEST model uses an empirical simplification: the discharge volume for urban areas is calculated using a two-dimensional model based on the runoff curve number (RCN) on saturated soils. The runoff curve number is a parameter that assumes that the volume of runoff will be highest where there is a highly sealed surface and where the soil has low conductivity. In areas where runoff occurs, the rainfall is not retained by the soil and flows into other parts of the city. In contrast, when sealed surfaces are fewer and soils have higher conductivity, the potential for stormwater retention is more significant.

Additionally, though we had the possibility to integrate our runoff analysis with the pluvial discharge system, we could not compare our modeling results with the digital extension of the sewage network. What we know is that in, the larger majority of the city, there is no difference between the two networks (rainwater and blackwater) thus it was impossible to empirically separate the quantity of runoff intercepted by the sewage system and the quantity gathered by the rainwater discharge system.

Lastly, this research has been developed during a preliminary phase for the preparation of a city's water management masterplan. There was no possibility of extending our assessment to potential long-term scenarios or alternative transformations since the design process was not included in our research activity. Therefore, even though this analysis serves as a basis to support the decision-making process, there is not an empirical evaluation regarding how the proposed NBS can reduce the vulnerability of the catchment. To cope with that issue, we tried to simulate a potential masterplan of transformation for a small portion of the Karsiyaka neighborhood while empirically checking the effective potential benefits of NBS [22] but we could not replicate this approach across the entire metropolitan area.

## 5. Conclusions

When planning the expansion of a city into upstream districts on sloped terrain at the urban fringe, it is critical to design sufficient infrastructure to manage stormwater in the upstream areas. Factors such as degree of slope, soil types, and percentage of surface sealing should be carefully considered when estimating stormwater runoff volume.

Within this work, we intended to create a geodatabase where the vulnerability of urban areas to cloudburst events was spatially mapped. To do that, we employed the InVEST urban flood risk mitigation model, which generated a map where the retention and flood volume after a 70 mm cloudburst event was simulated. This result has been used (i) to verify whether urban streams are sized enough to convey and discharge the potential runoff volume and (ii) to create a composite index of vulnerability made by the spatial overlap between the runoff index and the flow accumulation.

The analysis of vulnerability suggested that existing downstream areas may be greatly impacted by runoff originating in upstream areas within the same watershed. Urban areas that are expanding by the construction of new districts on surrounding slopes are especially vulnerable to cloudburst events because of an underestimation of the runoff volume that will originate upstream. The design of artificial water-carrying channels within urban areas must be based on an accurate estimate of the volume of runoff originating in new upstream areas.

The city of Izmir is a clear example of this phenomenon. Between 2000 and 2020, many highly sealed urbanizations have been built on the slopes around the city, which has resulted in increased and more frequent flooding of the city center on the plain below during rain events. To address this problem, which directly impacts a large proportion of the population of Izmir, nature-Based solutions such as detention ponds, the creation of green corridors, floodable parks, bioswales, rain gardens, water capture, and soil improvement are economical and ecological alternatives to traditional gray engineering solutions to

stormwater management. Nature-based solutions also provide social and recreational spaces, increase biodiversity in urban areas, and reduce the urban heat island effect.

**Supplementary Materials:** The following supporting information can be downloaded at: https://www.mdpi.com/article/10.3390/su142416418/s1, Geotiff S1: Hydrological Soil Groups.

**Author Contributions:** Conceptualization, K.V. and A.B.; methodology, S.S. and T.U.; formal analysis, S.S. and T.U.; investigation, S.S., N.S. and V.T.C.; resources, K.V., A.B. and T.U.; data curation S.S. and T.U.; writing—original draft preparation, S.S.; writing—review and editing, S.S., N.S. and V.T.C.; supervision, K.V.; project administration, A.B. All authors have read and agreed to the published version of the manuscript.

**Funding:** This research received no external funding.

**Institutional Review Board Statement:** Not applicable.

**Informed Consent Statement:** Not applicable.

**Data Availability Statement:** Not applicable.

**Conflicts of Interest:** The authors declare no conflict of interest.

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
