# Peer review of "Adapting Cities to Pluvial Flooding: The Case of Izmir (Türkiye)"

_sustainability, doi:10.3390/su142416418_

Round 1

Reviewer 1 Report

The topics of the article address an important issue related to the risks of urban areas in the context of observed environmental changes. I recommend the article for publication after taking into account the following comments.

What is the number of water bodies in the area under study. In Table 5 there is only a description of Water. What is the total retention capacity of the existing reservoirs/ponds? How this data was taken into account in the modeling performed.

Urbanized areas, due to extensive human interference, are quite difficult to contextualize. The paper lacks detailed information on the sewage network, which, despite the sealing of the ground, converts a large part of surface runoff into artificial underground runoff. What is the density of this network? What is its capacity? How does it change the volume of surface runoff?

Both situations, that is, reservoir retention and the transformation of surface runoff artificial underground runoff, are crucial in the context of the research being conducted.

Figure 4: What do the yellow and green colors mean? No legend.

Author Response

The topics of the article address an important issue related to the risks of urban areas in the context of observed environmental changes. I recommend the article for publication after taking into account the following comments.

What is the number of water bodies in the area under study. In Table 5 there is only a description of Water. What is the total retention capacity of the existing reservoirs/ponds? How this data was taken into account in the modeling performed.

Thank you so much for this comment. We thought these data were somehow redundant but we realized that maybe it would be much better to show more analytically the discharge capacity of the existing reservoirs and the potential flood volume predicted by the model. For this reason, we integrated the chapter “3.1. Runoff analysis of each sub-basin” while adding a new table.

Urbanized areas, due to extensive human interference, are quite difficult to contextualize. The paper lacks detailed information on the sewage network, which, despite the sealing of the ground, converts a large part of surface runoff into artificial underground runoff. What is the density of this network? What is its capacity? How does it change the volume of surface runoff?

Both situations, that is, reservoir retention and the transformation of surface runoff artificial underground runoff, are crucial in the context of the research being conducted.

Thank you so much for this comment. Unfortunately, even though we had the possibility to integrate our runoff analysis with the pluvial discharge system, we couldn’t have any digital data on the sewage network. What we know is that in the largest majority of the city there is no difference between the two networks (rainwater and blackwater) thus it was impossible to empirically separate the quantity of runoff intercepted by the sewage system and the quantity gathered by the rainwater discharge system.

Nevertheless, we took the opportunity to emphasize this as one of the limits of this study in our chapter “4.3 Limits and potentialities”.

Figure 4: What do the yellow and green colors mean? No legend.

Thank you so much for your comment, we re-edited and exported the map while enlarging the legend

Reviewer 2 Report

This well-written document piques my curiosity greatly. The innovative "nature-based solution" approach is perfect for a publication on sustainability.

Minor alterations

Check the unit format on lines 263-266.

Improved resolution is required for the figure because the letters are too small and illegible.

The data for the input and output of the InVEST model, as well as extra information, are needed.

Author Response

This well-written document piques my curiosity greatly. The innovative "nature-based solution" approach is perfect for a publication on sustainability.

Thank you so much for this positive comment.

Minor alterations

Check the unit format on lines 263-266.

Thank you so much for this observation, we corrected the units.

Improved resolution is required for the figure because the letters are too small and illegible.

Thank you for this comment, we changed both figure 2 and 3 while enlarging the legend and exporting the layout at 300 dpi.

The data for the input and output of the InVEST model, as well as extra information, are needed.

Thank you for your observation. Even though table 1 and table 3 are basically the input variables of the model, we will upload as supplementary material the original geotiff of the modelling session.